# Treatments and interventions addressing chronic somatic pain in torture survivors: A systematic review

**Tanzilya Oren**[1]*, **Nihan Ercanli**[2], **Omri Maayan**[1], **Samantha Tham**[1], **Drew Wright**[3], **Gunisha Kaur**[1]

**1** Department of Anesthesiology, Human Rights Impact Lab, Weill Cornell Medical College, New York, New York, United States of America, **2** College of Human Ecology, Cornell University, Ithaca, New York, United States of America, **3** Samuel J. Wood Library & C.V. Starr Biomedical Information Center, Weill Cornell Medical College, New York, New York, United States of America

* toren@fordham.edu

**Data Availability Statement:** The search strategy for this systematic review and all results are reported in the manuscript. All relevant data for this

## Abstract

Torture survivors experience chronic, somatic pain that may be exacerbated by environmental, social, and structural factors that extend beyond immediate traumatic events and diagnoses. We conducted a systematic review of research describing the types and efficacy of treatments for chronic somatic pain in a global population of torture survivors. In this systematic review, we searched Ovid MEDLINE, Ovid EMBASE (1974 to present), and PubMed. We used all appropriate controlled vocabulary and keywords for interventions and treatments for chronic somatic pain in torture survivors. The population included survivors of torture of any age and in any country. Outcomes included pain relief, pain intensity, distress level, and quality of life. Four authors participated in screening, full-text review, and quality assessment, with each title and abstract being independently reviewed by two authors. This study is reported according to the PRISMA guidelines and registered in PROSPERO. We included six pre-post intervention studies and four pilot or modified randomized controlled trials (RCTs), for a total of ten studies included in the analysis. Different combinations of interventions targeted pain reduction in refugees, the majority of whom were torture survivors as the primary (n = 1) or secondary (n = 9) outcome. Sample sizes varied from eight to 470 participants. We identified three main types of interventions: multimodal combined, manual therapy, and specific types of talk therapy. Five studies demonstrated positive outcomes on pain and its intensity, three reported no effect, and two had mixed outcomes. Pain in torture survivors is often considered a symptom secondary to mental health illness and not targeted directly. Instead, combined interventions are mainly directed at posttraumatic stress disorder (PTSD), depression, and anxiety. Most studies noted promising preliminary results and plans to conduct RCTs to increase the reproducibility and quality of their pilot data.

## Introduction

The United Nations Convention Against Torture (UNCAT) defines torture as the intentional infliction of extreme mental or bodily pain or suffering by state officials, with or without their

study are available within the paper and its Supporting information files.

**Funding:** National Institute of Neurological Disorders and Stroke Grant K23NS116114 (GK). The study funder, the National Institutes of Health, had no role in the conceptualization, data extraction, data analysis, or interpretation of the findings. All authors had access to the data in the study and accepted responsibility for the decision to submit the manuscript for publication.

**Competing interests:** The authors have declared that no competing interests exist.

knowledge, for a defined purpose [1]. The World Medical Association's broader definition of torture does not explicitly define perpetrators as government authorities or agents acting on their behalf; instead, it focuses on the coercive intent of any perpetrators to cause physical and psychological suffering [2].

Torture is associated with a wide range of chronic health conditions [3,4]. Pain and pain-related disabilities are the most common physical sequelae [5,6]. For example, *falanga*, or foot whipping, common in Asia, causes compensated gait, dysesthesia, and allodynia [7]; *ghotna*, or roller crushing of muscles, common in South Asia, causes pain on ambulation and severe quadriceps and adductor muscle pain [8]; *strappado*, or upper extremity suspension, common in the Middle East, results in brachial plexus injury, neuropathic pain, and complex regional pain syndrome [9]. The published literature on torture survivors suggests a high prevalence of chronic pain, with estimates ranging from 78% to 83% [10–14]. The pain experience of torture survivors is complex. This is in part due to compounded trauma: experiences of war and violence; layered with acute and chronic medical conditions; layered with migration-associated trauma such as family separation, immigration detention, and deportation stress, which are common in refugee torture survivors. While the symptom burden of pain in torture survivors is significant compared to other groups with chronic pain [15–17], it can also be difficult to diagnose, given variable cultural beliefs and expressions about pain [18], inability of torture survivors to access healthcare [19], and confounding illness such as posttraumatic stress disorder (PTSD) that often eclipse the physical sequelae of torture [20]. However, emerging research demonstrates that it is possible to diagnose chronic somatic pain in torture survivors, even from a wide range of countries, to the accuracy that approaches a physical exam by a specialist pain physician [14].

Once diagnosed, several interventions have been utilized in torture survivors—such as complex manual physical therapy, Cognitive Behavioral Therapy (CBT), and Narrative Exposure Therapy (NET)—but to variable effect. There are no large-scale, rigorous studies on the treatment options and their efficacies for chronic somatic pain in torture survivors, and clinicians' understanding of torture-induced chronic somatic pain and effective treatments is lacking [21–23]. Research primarily focuses on mental health illness in this population [24–27]. However, tested mental health interventions have not been shown to be effective for pain reduction. Further research is crucial for advancing theory development and enhancing the efficacy of available therapies [28,29].

The existing evidence for the treatment of pain after torture is meagre, generally of low quality due to very small sample sizes, and primarily focuses on psychological diagnoses and interventions. Only one prior systematic review investigated interventions for treating persistent pain in torture survivors [30]. Nearly seven years have passed since that review, which only evaluated three small RCTs, one of which was retracted. The objective of this review is to comprehensively identify, synthesize, and assess interventions and treatments targeting chronic somatic pain after torture.

## Methods

### Search strategy and selection criteria

This is a systematic review with a narrative summary of results conducted in accordance with PRISMA guidelines [31]. The protocol for this review was registered in PROSPERO (CRD42023409076). https://www.crd.york.ac.uk/prospero/display_record.php?RecordID= 409076. The PRISMA checklist is included in S1 Checklist.

## Search strategy and selection criteria

In collaboration with a medical librarian, we performed comprehensive systematic searches to identify studies that investigated torture and pain. Searches were run in April 2023 on the following databases: Ovid MEDLINE (In-Process & Other Non-Indexed Citations and Ovid MEDLINE 1946 to Present), Ovid EMBASE (1974 to present), and PubMed (all dates). The search strategy included all appropriate controlled vocabulary and keywords for torture and pain. The reference lists of the initially included papers were then searched using Scopus, with any new articles undergoing the same screening process. Medical Subject Headings (MeSH), Health Sciences Descriptors, and EMBASE Subject Headings (Emtree) were included in the search strategy. The search terms encompassed three main domains: torture survivors, treatment interventions, and pain outcomes. The database search strategy is attached in S1 Text. There were no language or publication date restrictions. Article types were limited to prospective intervention-testing studies such as RCTs, cross-sectional, case-control, cohort, mixed-method, and qualitative studies. Each abstract was screened for appropriateness by at least two independent reviewers, with a third resolving conflicts. The full texts of the abstracts deemed appropriate were then screened for inclusion by two independent reviewers, with a third resolving conflicts. Disagreements not resolved by the third reviewer were discussed between the four reviewers, and decisions were made by consensus. The stage-by-stage selection process was recorded in Covidence [32].

## Study selection

Prospective, clinical, and empirical studies testing specific interventions were included. If the interventions targeted chronic somatic pain exclusively or as a part of combination interventions targeting mental health in survivors of torture, they were included. The searches did not restrict by age of torture survivors, the country in which the study was conducted, or study sites. The studies were excluded if over 50% of participants were not torture survivors, did not test any treatment or intervention targeting pain, or did not address pain-related outcomes, distress levels, or quality of life as outcomes. Studies that were not in English or did not test any treatment or intervention were also excluded. The study selection is presented in PRISMA flow chart (Fig 1).

## Data analysis

Duplicate records were removed in Covidence. Four reviewers divided the remaining studies for extraction after title/abstract and full-text screening using our pre-set exclusion criteria.

The primary outcomes included pain relief, pain intensity, distress level, and quality of life. A systematic narrative synthesis was performed due to the expected heterogeneity of the research designs and methodology, with the date for the reported outcomes summarized in table format. Effect size estimates proved to be inappropriate based on the nature of the included studies.

Two reviewers performed a risk of bias assessment to analyze the quality of the final set of included studies using the Evidence Project risk of bias tool [33]. This tool is utilized to assess RCTs, non-RCTs, observational, and quasi-experimental studies. The risk of bias completed tool for this review is in S1 Table.

Data extracted included: characteristics of studies, intervention types and descriptions, rationales and outcomes, primary and secondary outcomes, results with specific metrics, measurement tools and instruments, method of aggregation, the aims and results of the studies, study participants' demographic information, and participants' retention data. These data

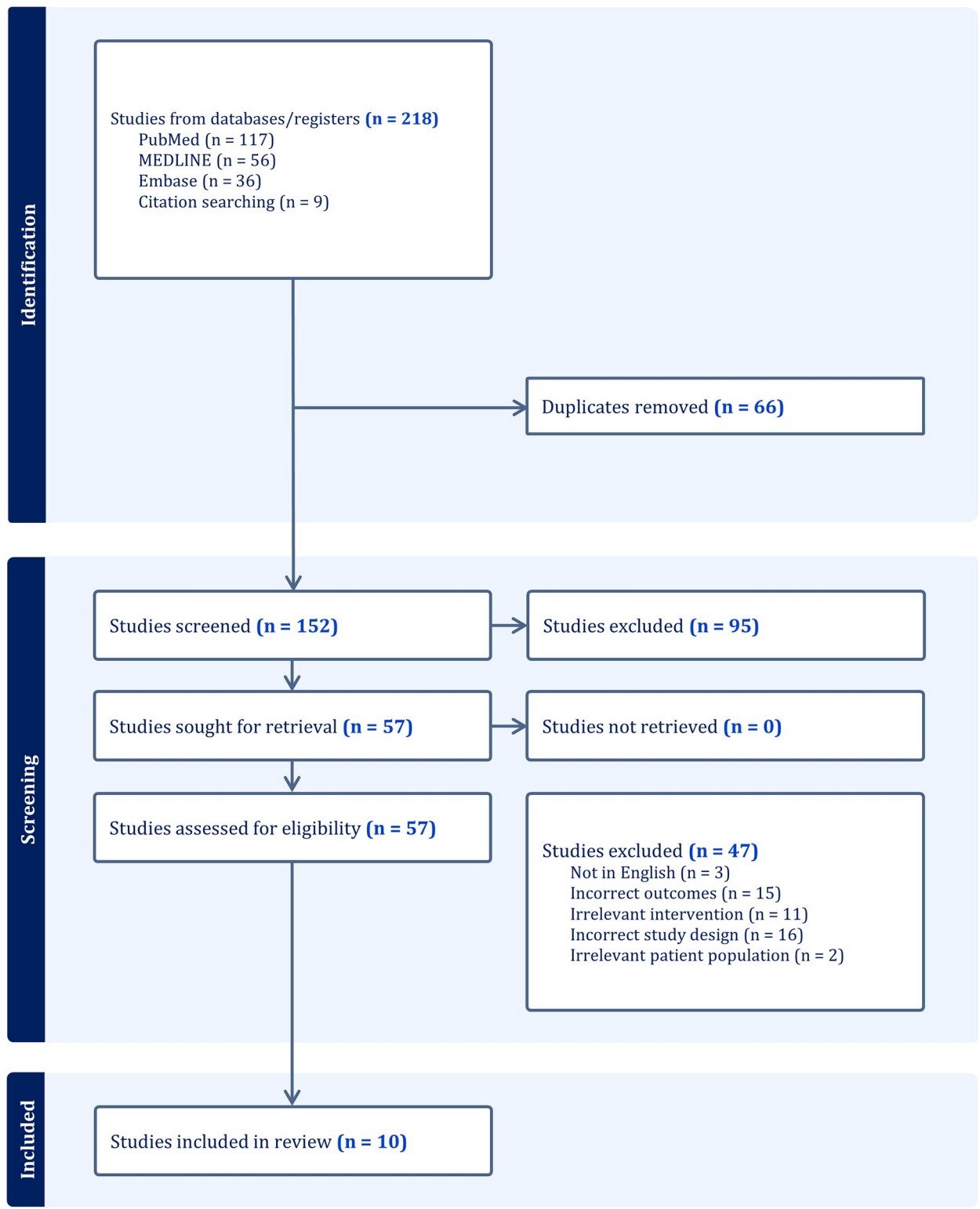

**Fig 1. PRISMA flow diagram for study selection.**

were assessed and recorded by three reviewers, with the fourth reviewer resolving any discrepancies.

Using a narrative text-based approach, a summary column was generated for each criterion to consolidate the numerical data and succinctly outline the similarities and distinctions observed across studies. All study outcomes were extracted, and all post-intervention changes in pain symptoms, distress levels, or quality of life were recorded. Due to the inclusion of highly heterogenous studies with varied measurements of outcomes, we performed a narrative systematic review with basic study descriptions and a summary of intervention results in table form.

## Results

The searches yielded 218 studies, with 66 duplicates removed. We screened the titles and abstracts of 152 studies and removed 95 unrelated studies. The full texts of the remaining 57 studies were screened closely against inclusion and exclusion criteria. Ten studies were included in the analysis after excluding 47, which did not meet the pre-set criteria, such as studies not published in English; outcomes not mentioning pain, distress, or quality of life; study designs not testing an intervention; and samples not delineating the proportion of torture survivors. A total of ten studies were ultimately included. The PRISMA flowchart is presented in Fig 1.

The included studies were conducted in ten different countries, including one in North America, four in Europe, four in Asia, and one in Africa (Table 1). The populations included ten different ethnic groups representing a global population of refugees. All ten studies entailed pre-post testing of an intervention. There were four pilot or pragmatic RCTs (three pilot RCTs and one pragmatic parallel-group one) [34–37], two treatment follow-up studies [38,39], three pre and post-test quasi-experimental studies [40–42], and one A-B design study [43]. Interventions included complex multidisciplinary approaches targeting pain, mental health, and well-being (multimodal interventions), with two studies using singular interventions such as Narrative Exposure Therapy (talk therapy) and complex manual therapy (physical therapy).

The tools and instruments used to measure changes in pain, pain intensity, related distress levels, and quality of life outcomes included part C of the Composite International Diagnostic Interview (CIDI-C) of the World Health Organization, a generic body diagram to pinpoint pain locations, short form of the Brief Pain Inventory (BPI), full version of the BPI, internally developed 5-item Pain Scale, short form of McGill Pain Questionnaire (SF-MPQ), Wong-Baker FACES Pain Rating Scale, World Health Organization Disability Assessment Schedule 2.0 (12 items) (WHODAS 2.0), Norwegian Pain Association's Minimum Inventory for Pain Patients (NOSF-MISS), and Social Participation scale (P-scale, Pain and Anger Analogues).

Sample sizes varied from eight to 82 in smaller studies and from 214 to 470 in larger ones, with 1,216 participants recruited in total. The gender distribution was roughly equal, with 575 (48%) female and 626 (52%) male participants retained across the ten included studies. Two studies had only male study participants. Nine studies reported that the intervention worked for treating complex mental and physical sequelae of torture, while one had inconclusive results. The general limitation of all studies was a small number of participants (in six studies), nonrandomization of participants at the selection stage, and the experimental, pilot-like nature of all studies.

### Definitions of torture and pain in the studies' populations

The inclusion of individuals who have undergone torture was a key criterion in the selection process for most participants in the included studies. Given that torture is not considered a

**Table 1. Summary of characteristics of included studies.**

| Study | Country where study conducted | Study design | Population | No of participants at baseline | No of survivors of torture at baseline | Mean age (range, SD), total or control; intervention | Ethnic groups or country of origin of participants | Intervention type(s) | Outcome domains |
|---|---|---|---|---|---|---|---|---|---|
| **Dibaj 2017** [43] | Norway | Pre-post | Refugees-torture survivors at the clinic | 8 | 8 | NM, 30s–60s, NSD | Middle East (3), the Caucasus (2), Central Africa (1) | Narrative Exposure Therapy (NET) and physiotherapy | PTSD and pain |
| **Dix-Peek 2018** [42] | South Africa | Pre-post | Individuals who have been affected by torture at the clinic | 82 | 82 | 34.82 (18–72; 8.68); 36.20 (18–72, 10.35) | Burundi (3), DRC (28), Eritrean (4), Ethiopian (21), Somali (15), South African (4), Zimbabwean (2), Other (5) | Multimodal framework: aspects of trauma-focused CBT (TFCBT), Narrative Exposure Therapy (NET), dialectical behavioral therapy, supportive therapy, problem-solving and solution-focused therapy | PTSD, anxiety, depression, pain and social functioning |
| **Gamble 2020** [34] | Iraq | Pilot RCT | Incarcerated male survivors of torture | 30 | 30 | 33.2 (NR, NSD) | Kurdish | Physiotherapy and psychotherapy | Pain, anxiety, depression, PTSD, sleep, physical functioning and self-efficacy |
| **Jorgensen 2015** [38] | India | Pre-post | Survivors of torture and ill-treatment in the community and at the clinic | 470 | 357 (76%) primary survivors 113 (23%) were secondary survivors | NM (15–80; NSD) | Indians (different castes and religions) | Testimonial Therapy | Well-being, social participation, pain and anger |
| **Kim 2015** [41] | South Korea | Pre-post | Survivors of torture with low back pain who were patients of a clinic | 30 | 30 | 62.6 (NR; 6.6); 59.2 (NR; 6.6) | South Korean | Complex manual therapy | PTSD, pain, and lumbar function |
| **Neuner 2010** [37] | Germany | Pilot RCT | Survivors of physical torture and other traumatic events at the clinic | 32 | 28 | 31.6 (NR; 7.7); 31.1 (NR; 7.80) | Turkey (25), Balkans (4), Africa (3) | Narrative Exposure Therapy (NET) | PTSD, depression, pain |

*(Continued)*

**Table 1.** (Continued)

| Study | Country where study conducted | Study design | Population | No of participants at baseline | No of survivors of torture at baseline | Mean age (range, SD), total or control; intervention | Ethnic groups or country of origin of participants | Intervention type(s) | Outcome domains |
|---|---|---|---|---|---|---|---|---|---|
| **Nordin 2019** [39] | Denmark | Pre-post | Tortured refugee patients at the clinic | 276 | 226 (82% had been subjected to torture), 50 subjected to other forms of organized violence | 44.8 (NR; 9.4) | Iraq (38%), Iran (15%), Lebanon (95), Bosnia (6%), and Afghanistan (5%). Also, Somalia, Syria, Egypt, Russia, and Turkey. | Multidisciplinary therapy (trauma-focused psychotherapy, strategies to cope with pain/somatic difficulties, physical exercise routines, training in body awareness and relaxation exercises, management of pain, sleep, and psychotropic medications, sessions addressing social difficulties and integration into social network/society). | PTSD, depression, anxiety, pain, disability |
| **Northwood 2020** [36] | United States | Pragmatic RCT | Karen refugee patients exposed to war and torture at the clinic | 214 | 77 reported torture, 144 reported direct harm | 42.76 (18–65; 3.28) | Karen from Burma | Narrative Exposure Therapy, Cognitive Behavior Therapy, Sensorimotor Psychotherapy, and patient-centered methods such as Motivational Interviewing; Case Management | Depression, anxiety, pain, PTSD and social functioning |
| **Phaneth 2014** [40] | Cambodia | Pre-post | Torture rehabilitation patients at the clinic | 40 | 40 | 52 (31–72; 11) | Cambodian | "Pain school" (Ten-session, group-based, interdisciplinary pain education intervention) | Disability and pain |
| **Wang 2016** [35] | Kosovo | Pilot RCT | Victims of torture and war in Kosovo at the clinic | 34 | 34 | 48.8 (NR; 10.9); 46.8 (NR; 10.4) | Northern Kosovars | Cognitive behavioral therapy (CBT), breathing exercise with an emWave biofeedback device, and group physiotherapy | Mental (PTSD, depression, anxiety), emotional (anger, aggressiveness, inferiority complex, social isolation, and police or military phobia), and physical (chronic pain symptoms, body mass index, handgrip strength, and standing balance) health, and social outcomes (income, employment rate, and disability score) |

NM = No Mean; NSD = No standard deviation; NR = No range.

clinical diagnosis in and of itself, but rather a self-reported traumatic experience with legal and political implications, there are a limited array of instruments available to systematically ascertain the extent of traumatic events identified as torture. Three studies adopted the definition of torture outlined in the United Nations Convention Against Torture (UNCAT), while also incorporating additional torture-like experiences. Three other studies did not provide explicit definitions of torture, and incorporated organized violence and violence associated with war as part of their inclusion criteria. Four studies did not offer any explicit definition of torture. In three studies, victimization by organized violence, direct harm from exposure to war and torture, sexual harassment, molestation, rape, and other cruel, inhuman, or degrading treatment—criteria which are similar to the UNCAT definition—constituted participant inclusion criteria. These inclusion criteria were applied in addition to, or as a substitute for, self-reported torture exposure.

In addition to torture experiences, a history of chronic somatic pain was the main clinical inclusion criteria in two studies, PTSD and other psychiatric and affective disorders were the main clinical criteria in three studies, both pain and affective disorders were included in three studies, and two studies included pain and mental health disorders but did not specify any clinical diagnoses as inclusion criteria. In five studies where pain was an explicit inclusion criterion, it was defined as somatic symptoms, chronic non-malignant pain, comorbid chronic pain, low back pain, or persistent neuropathic and nociplastic pain.

## Randomized control trials

Three trials were conducted at clinics serving refugees in Europe or the United States, and one trial was conducted in an Iraqi prison (Table 1). In total, the four trials included 114 men and 196 women, with one trial only including men [34]. The ages of participants ranged from 18 to 65 years, and three studies reported an average age range of 31–48 years [34,35,37].

The types of traumatic events reported by the study population included physical assault, verbal abuse, solitary confinement, torture, sexual harassment, assault, and witnessing an assault on a familiar person [34,35,37]. One trial did not report the types of torture experienced by participants [36]. All trials excluded participants if they had an acute or confounding psychiatric condition (e.g., psychosis) and/or were high risk to self or others. Three trials excluded participants if they reported acute mental health issues with current use of mental health services [34–36]. The list of studies excluded at the full-text review stage are presented in S2 Table.

Changes in PTSD symptoms were also assessed across all studies. Two studies used the Post-traumatic Diagnostic Scale [36,37] and two studies used the Harvard Trauma Questionnaire [34,35].

As for outcomes, three studies reported significant reductions in pain symptoms as well as posttraumatic stress in participants following treatment [34,36,37], though one study noted that the significant improvement of pain was not statistically significant after adjusting for the interaction of tests between pain and depression [37]. One study reported inconsistent patterns in the chronic pain outcome [35].

## Pre-post-test studies

Six studies can be generally categorized as pre-post-test studies. Three studies collected data at baseline and at a singular time-point post-intervention [38,40,41], while the remaining three studies collected data for at least two time-points in addition to baseline data [39,42,43]. Five studies used baseline data in their analysis of the impact of the interventions, and one study used a three-month waiting list condition as a comparator for the initial

treatment group [42]. The duration of each intervention exposure episode was similar for all six studies, with the sessions of the different interventions lasting from one hour to two hours. The length of participant follow-up varied from one month [38] to nine months after the last session [39].

Regarding treatments, two of the pre-post-test studies utilized Narrative Exposure Therapy (NET) in some capacity within the intervention [42,43]. One of these studies also utilized physiotherapy [43], while the other also utilized a wider multi-modal framework with trauma-focused CBT (TFCBT) and other broader therapies [42]. The remaining four studies are otherwise heterogeneous in their intervention types, with interventions consisting of Testimonial Therapy (TT), complex manual therapy, multi-disciplinary therapy, and "pain school" [38,39,41]. All six of these studies looked at pain in some capacity within outcome domains. Four studies included PTSD as one of their outcome domains [39,41–43], two studies included depression and anxiety in their outcome domains [39,42], and two studies included disability in their outcome domains [39,40]. Three studies had a variety of other outcome domains such as social functioning, social participation, and lumbar function [38,41,42].

For outcome metrics measuring pain, three studies used the Brief Pain Inventory (BPI), allowing patients to rate the severity and degree to which the pain interferes with well-being [39,40,43]. One study also used the Norwegian Pain Association's Minimum Inventory for Pain Patients (NOSF-MISS) to assess pain intensity [43].

Two studies assessed the number of areas of pain, with one study using body diagrams to mark areas of pain [39] and another using verbal identification of pain [42]. Two studies used scales for pain intensity, one study used a pain analogue Likert scale from 0 to 5 [38], and one study used the Visual Analog Scale (VAS) to identify pain scores directly on a scale of 0 to 10 [41].

All studies saw a statistically significant change in at least one of the outcome domains measured, and concluded that the interventions were successful. Regarding pain outcomes specifically, the heterogeneity of the specific metrics limits the comparisons that can be made. However, significant changes were exhibited for each of the respective pain metrics. Three studies found significant pre-to-post-treatment reductions in pain using various measurements: number of pain locations [39], mean pain in the last 24 hours [40], and areas of pain [42]. One study found that two of eight participants had a significant reduction in pain intensity [43]. Two studies found significant improvements in their chosen pain metrics: self-perceived pain [38] and the VAS for pain [41]. Among domains other than pain outcomes, the studies found varying levels of significance, and all studies concluded that their respective interventions were generally successful in improving participant outcomes.

In summary, the evidence for treatment and intervention effectiveness for chronic pain is mixed. The ten selected studies tested various treatments and interventions that targeted pain as the main (in one study) or secondary outcome (in nine studies) in refugees, the majority of whom were torture survivors. The measurement tools used to assess the outcomes were highly heterogeneous. The three identified main types of interventions—multimodal (multidisciplinary or interdisciplinary) combined, manual therapy alone, and specific types of talk therapy alone—generally worked for reducing the symptoms of PTSD, anxiety, and other mental health conditions, except for one inconclusive study. As for chronic pain outcomes, five studies demonstrated positive outcomes on pain and its intensity, while three reported no effect on pain, and two had mixed outcomes for pain. The characteristics of the interventions, treatments, and outcomes are reported in Table 2.

**Table 2.** Intervention description and overall effect on pain outcomes.

| First author and year | Intervention and its description or components | Control Group | Did the intervention work for pain? |
|---|---|---|---|
| **Dibaj 2017** [43] | Combination of NET and physiotherapy | No control group | Mixed. Two patients had decreased pain intensity, two had no change and one experienced increased pain. One patient achieved a clinically significant decrease in pain intensity. |
| **Dix-Peek 2018** [42] | Centre for the Study of Violence and Reconciliation (CSVR) framework Aspects of trauma-focused CBT (TFCBT), NET, dialectical behavioral therapy, supportive therapy, problem-solving and solution-focused therapy underpin therapeutic interventions in the framework, with an emphasis on empowerment. | Intervention vs. Waitlist control group | No. Inconclusive evidence, but there was a general trend of improvement in the psychological well-being and functioning of both groups. |
| **Gamble 2020** [34] | Interdisciplinary: Physiotherapy and psychotherapy The physiotherapy group treatment included relaxation exercises, mindfulness exercises, breathing exercises, stretching and strengthening exercises, low to moderate-intensity exercise, therapeutic neuroscience education, circuit training, body awareness exercises, and interactive education regarding coping skills, sleep, mind-body connection, etc. Psychotherapy group treatment included stabilization techniques and coping skills, breathing and mindfulness exercises, psychoeducation, techniques based on dance movement therapy and somatic psychology, techniques based on CBT, strategies for reflecting on loss and grief, goal setting, and planning for the future. | Intervention vs. Waitlist control group | Yes. Statistically significant improvement in all measures, including anxiety and/or depression, PTSD, and nociplastic pain. |
| **Jorgensen 2015** [38] | Testimonial Therapy (TT) In TT a torture survivor, a note taker, and a therapist produce a written testimony about the human rights violations, which the survivor has suffered. Then a village ceremony is held, where the testimony is read out loud to the audience by the therapist or the survivor. After the presentation of the testimony, the survivor receives a garland to symbolize a transition (rite of passage) from victim to survivor. | No control group | Yes. Statistically significant improvements in the Pain Analogue (pre-therapy average: 3.1 post-therapy average: 0.94) |
| **Kim 2015** [41] | Complex Manual Therapy Includes Myo-Facial Release (MFR), the Muscle Energy Technique (MET), pelvic posterior tilt exercise; upper abdominal exercises; lumbar stabilization exercise extension exercise for muscle strength by bridge exercise a with sling, and self-exercise | Intervention vs. No treatment control group | Yes. Visual Analogue Scale (VAS) results after complex manual therapy were significantly lower in the experimental group than in the control group. |
| **Neuner 2010** [37] | NET NET focuses on the entire survivor's history, including all traumatized events, rather than a particular event for therapy. A neurocognitive memory theory underpins NET, which predicts that completing autobiographical memories of traumatic events and connecting them to fear memories is one of the key agents of change in trauma therapy. | Intervention vs. Care-as-usual control group | No. The effect of NET on pain was not significant. |
| **Nordin 2019** [39] | Multidisciplinary therapy Trauma-focused psychotherapy, strategies to cope with pain and somatic difficulties, physical exercise routines, training in body awareness and relaxation exercises, management of pain, sleep, and psychotropic medications, sessions addressing social difficulties, and integration into social network/society. | Intervention vs. Waitlist control group vs. Intervention-completed group | Mixed. Treatment results were statistically significant for a reduction in PTSD, depression, anxiety, and a number of pain locations. But no reductions were observed in pain severity, and health-related disability, except for societal participation. |

*(Continued)*

**Table 2.** (Continued)

| First author and year | Intervention and its description or components | Control Group | Did the intervention work for pain? |
|---|---|---|---|
| **Northwood 2020** [36] | Combination of NET, CBT, Sensorimotor Psychotherapy, and patient-centered methods (Motivational Interviewing and Case Management). | Intervention vs. Care-as-usual control group | Yes.<br>Results were statistically significant for reductions in depression, anxiety, PTSD, and pain symptoms from baseline to 3 months. Positive treatment effects continued through 12 months in all symptom outcomes for the Intensive Psychotherapy and Case Management (IPCM) group. The between-groups difference was significant at 12 months. |
| **Phaneth 2014** [40] | "Pain school"<br>Ten-session, group-based, interdisciplinary pain education intervention. Each session unit taught about pain or intervention (i.e., "pain mechanisms" included two sessions in which pain-relevant anatomy was taught. Drawings and body models are used to explain such phenomena as "false alarm signals" associated with chronic benign pain versus "real" pain signals due to tissue damage with acute pain). | No control group | Yes.<br>The results for the items from the BPI and the additional item "average pain during the last 24 hours" were significant for the reduction of scores, except for sleep. |
| **Wang 2016** [35] | Multidisciplinary intervention: CBT with Biofeedback (BF) via heart rate variability (HRV) device, Prolonged Exposure Therapy (PET), group activities.<br>CBT interventions are based on an adaptation of PET, with exposure to trauma memories, psychoeducation, and anger management. Breathing re-training used a HRV biofeedback device.<br>Group sessions: A series of physical games and activities to enhance their physical activity and participation level. | Intervention vs. Waitlist control group | No.<br>Inconsistent patterns with mental health and chronic pain outcomes were observed. |

Abbreviations for common interventions are as follows: NET = Narrative Exposure Therapy; CBT = Cognitive Behavioral Therapy.

## Discussion

This systematic review of the types and efficacy of treatments for chronic somatic pain in survivors of torture worldwide includes ten intervention-based studies conducted in ten different countries. We identified three main types of interventions: multimodal combinatorial interventions, manual therapy, and types of talk therapy. Half of the studies reported positive outcomes on pain and pain intensity. The quality of included studies was generally good based on the Evidence Project Risk of Bias assessment, notwithstanding significant limitations such as lack of randomization and control groups in some studies, small sample sizes in six studies, and the pilot nature of most studies.

In the included studies, several interventions demonstrated promise, including physiotherapy/psychotherapy [34], Testimonial Therapy [38], Complex Manual Therapy [41], "Pain School" [40], and combined therapies [36]. It is notable that interventions that resulted in statistically significant improvement in pain measures, with the exception of one study [38], treated pain as a *somatic* (i.e., interventions included muscle strengthening, movement therapies, etc.) rather than as a purely *psychosomatic* illness. These data are in line with recent findings that document how chronic somatic pain after torture accords with mechanism of injury (e.g., brachial plexopathy after *strappado*; lumbosacral plexopathy after leg hyperextension) in a vast majority of cases [44]. Countering the assumption that pain after torture is an expression of PTSD, depression, anxiety, or somatization, this provides diagnostic and treatment opportunities to improve the rehabilitation of torture survivors.

There was vast heterogeneity in the instruments utilized for pain evaluation in the studies included in this systematic review. However, several, including the BPI, BPI short form, and

the VAS, demonstrated diagnostic utility. Other studies have shown that while torture survivors with chronic pain have complex clinical presentations, accurate diagnosis with standard, validated pain screening tools is possible, to an accuracy that approaches a physical examination of a specialist pain physician [14]. These kinds of screens are exceptionally important in an environment where the need for trauma-evaluation-trained clinicians who have advanced training to care for torture survivors greatly exceeds the global availability of such providers. Accurate screens allow for diagnosis by non-specialist providers.

Refugees and asylum seekers have been the main subjects of the research on torture and its sequelae. These populations frequently encounter experiences that result in their marginalization, leading to a lack of representation and influence in advocating for improved treatments and interventions. Further, there is often a hesitation to engage in research with extremely vulnerable populations, resulting in the ethical harm of exclusion of these patients from research. However, given the exponential rise of forcibly displaced refugee populations globally—currently at one in less than 100 people as reported by the United Nations High Commissioner for Refugees—of which 40% are survivors of torture, healthcare providers across the world will increasingly encounter and care for these patients [45,46]. This is a critical area of research, where evidence-based guidelines and approaches must be developed.

This study is limited by the small size and quality of the included studies. Further, the studies' populations and measurement tools used to assess the outcome of interest (pain intensity, related distress, and quality of life outcomes) were highly heterogeneous, preventing us from conducting a meta-analysis. Future studies should avoid these pitfalls by ensuring a globally representative patient population, large enough sample size to be powered for definitive results, and utilizing the most rigorous study structures (e.g., RCTs with control groups). Within our own study, risk of bias was reduced by the input of several independent reviewers, and utilization of standard quality assessment tools.

The diverse range of multimodal therapies implemented in the research included in this analysis demonstrates a high level of innovation and novelty in addressing both mental health and somatic pain. A primary takeaway from this investigation is that further research is necessary across several domains, due to the limited effectiveness observed in the evaluated therapies for chronic pain. In the included studies, many researchers discussed their intentions to expand and conduct RCTs to enhance the reproducibility and quality of the promising pilot data that they produced. Larger, definitive studies on physiotherapy/psychotherapy, Testimonial Therapy, Complex Manual Therapy, "Pain School," and combined therapies that consider pain as a manifestation of physical illness, should be designed and executed. Conducting such investigations is of paramount importance in advancing theoretical frameworks and optimizing existing therapeutic interventions to rehabilitate torture survivors.

## Supporting information

**S1 Checklist. S1 PRISMA checklist.**
(DOCX)

**S1 Table. Risk of bias assessments.**
(DOCX)

**S2 Table. Characteristics of excluded studies.**
(DOCX)

**S1 Text. Search strategy.**
(DOCX)

## Author Contributions

**Conceptualization:** Tanzilya Oren, Gunisha Kaur.

**Data curation:** Tanzilya Oren, Nihan Ercanli, Omri Maayan, Samantha Tham, Drew Wright.

**Formal analysis:** Tanzilya Oren, Nihan Ercanli, Omri Maayan, Samantha Tham.

**Funding acquisition:** Gunisha Kaur.

**Methodology:** Tanzilya Oren, Nihan Ercanli, Drew Wright.

**Resources:** Drew Wright.

**Software:** Nihan Ercanli, Omri Maayan, Drew Wright.

**Supervision:** Gunisha Kaur.

**Validation:** Nihan Ercanli, Samantha Tham.

**Visualization:** Drew Wright.

**Writing – original draft:** Tanzilya Oren, Omri Maayan.

**Writing – review & editing:** Nihan Ercanli, Omri Maayan, Samantha Tham, Gunisha Kaur.

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
