## [Decision Letter · Decision Letter 0]

30 Jan 2024

PGPH-D-23-02521

Treatments and interventions addressing chronic somatic pain in torture survivors: A systematic review

Dear Dr. Oren,

Thank you for submitting your manuscript to PLOS Global Public Health. After careful consideration, we feel that it has merit but does not fully meet PLOS Global Public Health’s publication criteria as it currently stands. Therefore, we invite you to submit a revised version of the manuscript that addresses the points raised during the review process.

We look forward to receiving your revised manuscript.

Kind regards,

Muhammad Asaduzzaman, MD MPH MPhil

Academic Editor

Journal Requirements:

Additional Editor Comments (if provided):

Reviewers' comments:

Reviewer's Responses to Questions

**Comments to the Author**

1. Does this manuscript meet PLOS Global Public Health’s publication criteria? Is the manuscript technically sound, and do the data support the conclusions? The manuscript must describe methodologically and ethically rigorous research with conclusions that are appropriately drawn based on the data presented.

Reviewer #1: Yes

Reviewer #2: Yes

Reviewer #3: Yes

2. Has the statistical analysis been performed appropriately and rigorously?

Reviewer #1: N/A

Reviewer #2: Yes

Reviewer #3: N/A

3. Have the authors made all data underlying the findings in their manuscript fully available (please refer to the Data Availability Statement at the start of the manuscript PDF file)?

Reviewer #1: Yes

Reviewer #2: Yes

Reviewer #3: Yes

4. Is the manuscript presented in an intelligible fashion and written in standard English?

Reviewer #1: Yes

Reviewer #2: Yes

Reviewer #3: Yes

5. Review Comments to the Author

Reviewer #1: The authors report the results of a harduous research, either for the scarce literature available on the issue "torture",or for the need to enphasize the consequences of such trauma, not only psychologically but also in terms of somatic pain, a topic likely underevaluated or rarely addressed. Papers not written in English were not taken into account, and this could be a fault of the paper. Nevertheless, the paper is well written and the results are properly analyzed. References are appropriate and numerous.

Limited effectiveness and variable outcomes were observed in evaluating therapies for chronic pain following tortures. Accordingly, this paper is a clear stimulus to further research in order to highlite the problem and optimise existing therapeutic interventions

Reviewer #2: I would like to extend my thanks for getting the chance to review this spectacular paper. I truly believe the paper has strong qualities, but i will focus on the areas for improvement. I have put forth my suggestions below for the consideration of the authors. I hope the you will address the suggestions before endorsement of publications, especially those in the discussion section

Introduction

The introduction section has a truly informative flow and you have justified the need for the study with out a doubt.

You have mentioned how the pain experience of the torture survivors is complex, if you could explain on that a little bit further in an additional paragraph, where you can also extend by outlining possible interventions and treatments currently on practice, irrespective of the evidence meagerness, you will definitely set up your readers in the best position.

Line 60 - The previous comment is best entertained after this paragraph

Methods

Search strategy and selection criteria

I truly believe you are addressing an area with critical evidence need. So Is it possible to consider meta-analytic approaches to quantify the effect size of the interventions?I believe, if that's is possible, Your review would be a milestone in pain management of torture survivors. if possible, please go for it , even though the studies may prove to be highly heterogeneous

Line 91 - its better to put your search strategy domains in standard PICO format to ensure clarity and comparison with future studies.

Study selection

In addition to assessing eligible studies, based on inclusion and criteria, through full text review, please also include if you have taken any quality assessments.

Line 106 - Since your study focused on pain after torture as clearly shown in you inclusion criteria, it will be prudent to discuss a little bit on peculiarities chronic pain in those with torture as compared to other common caused of chronic pain. This will further justify the need for considering pain uniquely in torture survivors

Data analysis

The previous comment on meta-analytic approaches is a kind plea for possible reconsideration. But please note, i fully accept your justifications.

Result

Definitions of torture and pain in the studies’ populations

Please discuss a little on this torture-like experiences on the introduction and their possible implication on the chronic pain experiences, this would best position your readers

Table 2. Intervention Description and Overall Effect on Pain Outcomes.

Its better to keep some kind of order to the list of studies here and in the previous descriptive table of studies, either sample size, alphabetically, or just keep the orders of the two tables similar.

I would also strongly suggest the inclusion of additional column to clearly put the different intervention and control groups, or just the intervention group for those with no control group. because i seem to get lost between which studies were randomized or not so better include it here in a clear manner just for simplicity

Northwood 2020 - IPCM? - Please write this acronym fully, since you used it for first time here

Dix-Peek 2018 - CSVR framework - also consider writing out this acronym

Discussion

I truly believe you could work on the depth of your discussion, considering the "dearth of knowledge" existing in this research question. I suggest few ideas for you to discuss on in this section;

* Discuss on the intervention approach which had consistently shown improvement of pain outcome, speculate about why the approach would have been effective compared to the others triangulating with the current pathophysiological or psychological understanding

*Also discuss on the possible impact of pain measurement tools, those used by most studies, sensitivity in detecting impact of interventions . discuss this with context of clinical practice and its implication for those treating this patients..

*Also sift targeted recommendations for practices on patient management, though inconclusively

*Also elaborate further on possible pitfalls to avoid on future study designs you have recommended on the last paragraph.

Reviewer #3: Overall, this is a methodologically rigorous systematic review of an important topic, i.e., interventions addressing chronic somatic pain in torture survivors. It is well researched and well written and should be accepted with minor revisions; focused mostly on presentation of the information contained within the second table. Specifically, I would suggest the following:

- Introduction is thin; would consider adding some more information about the most common modalities subsequently mentioned and their demonstrated efficacy in other special populations. Would also mention some of the challenges and comparative strategies for assessing/measuring pain including across cultures.

- Posttraumatic Stress Disorder should be spelled out at first mention with (PTSD) alongside, then referred to by abbreviated name moving forward.

- Page 9 lines 203-204: re: "All trials excluded participants if they presented with a psychiatric condition" - this is a bit confusing given the prevalence of PTSD in this population; is this referring to an acute psychiatric emergency?

- Page 13 lines 227-228: re: "two studies utilized Narrative Exposure Therapy (NET) in some capacity within the intervention" - this is confusing as NET is mentioned in four different interventions in Table 1 (Neuner, Northwood, Dibaj, and Dix-Peek)

- Table 2: I recommend splitting into two tables as follows:

-- One table should include definitions of all individual components/modalities mentioned (i.e., NET, CBT, Sensorimotor Psychotherapy, BF, PET, PT, motivational interviewing, case management, DBT, TT, MFR, etc.) independent of the studies. You could add a column mentioning which studies it was included in (e.g., for NET, Neuner, Northwood, Dibaj, and Dix-Peek) +/- the percentage of studies which employed it to get a sense of how commonly it was being used.

-- A separate second table should then describe exactly what each of the interventions did, engaging abbreviations from preceding table. For example, for Dibaj 2017, NET and PT would be defined in initial table whereas the second table would describe intervention as "20 sessions of NET (90 min each) and 10 sessions of PT (60 min each). For the second table, results should include which measures were being used (e.g., for Dibaj, NOSF-MISS).

- Page 17 lines 283-284: re: "the quality of included studies was generally good) - how did you make this assessment? Did you consider using a Mixed Methods Appraisal Tool (MMAT) to formally assess quality? https://www.mcgill.ca/familymed/research/projects/mmat

- Page 17 lines 292-294: re: "Given the exponential rise of forcibly displaced refugee populations – currently at one in ten people" - what is 1 in 10 referring to? Surely not overall number of displaced individuals worldwide (108 million, or approximately 1.35 per 100 total world population).

- Discussion is quite limited; would expand significantly and make clear what the take-home points should be for readers (consider adding Conclusions section).

6. PLOS authors have the option to publish the peer review history of their article (what does this mean?). If published, this will include your full peer review and any attached files.

**Do you want your identity to be public for this peer review?** For information about this choice, including consent withdrawal, please see our Privacy Policy.

Reviewer #1: No

Reviewer #2: **Yes: **Dr Henok Tadesse

Reviewer #3: No

---

## [Decision Letter · Decision Letter 1]

12 Mar 2024

Treatments and interventions addressing chronic somatic pain in torture survivors: A systematic review

PGPH-D-23-02521R1

Dear Tanzilya Oren,

We are pleased to inform you that your manuscript 'Treatments and interventions addressing chronic somatic pain in torture survivors: A systematic review' has been provisionally accepted for publication in PLOS Global Public Health.

Best regards,

Muhammad Asaduzzaman, MD MPH MPhil

Academic Editor

Reviewer Comments (if any, and for reference):

Reviewer's Responses to Questions

**Comments to the Author**

1. If the authors have adequately addressed your comments raised in a previous round of review and you feel that this manuscript is now acceptable for publication, you may indicate that here to bypass the “Comments to the Author” section, enter your conflict of interest statement in the “Confidential to Editor” section, and submit your "Accept" recommendation.

Reviewer #2: All comments have been addressed

Reviewer #3: All comments have been addressed

2. Does this manuscript meet PLOS Global Public Health’s publication criteria? Is the manuscript technically sound, and do the data support the conclusions? The manuscript must describe methodologically and ethically rigorous research with conclusions that are appropriately drawn based on the data presented.

Reviewer #2: Yes

Reviewer #3: Yes

3. Has the statistical analysis been performed appropriately and rigorously?

Reviewer #2: N/A

Reviewer #3: N/A

4. Have the authors made all data underlying the findings in their manuscript fully available (please refer to the Data Availability Statement at the start of the manuscript PDF file)?

Reviewer #2: Yes

Reviewer #3: Yes

5. Is the manuscript presented in an intelligible fashion and written in standard English?

Reviewer #2: Yes

Reviewer #3: Yes

6. Review Comments to the Author

Reviewer #2: Thank you for generously addressing my suggestions, I am honored to have reviewed such a landmark research article.

Reviewer #3: Excellent revision, no further comments. This will be a very valuable addition to the literature.

7. PLOS authors have the option to publish the peer review history of their article (what does this mean?). If published, this will include your full peer review and any attached files.

**Do you want your identity to be public for this peer review?** For information about this choice, including consent withdrawal, please see our Privacy Policy.

Reviewer #2: **Yes: **Dr Henok Tadesse Bireda

Reviewer #3: No
